# Age-Specific Normative Values of Lumbar Spine Trabecular Bone Score (TBS) in Taiwanese Men and Women

**DOI:** 10.3390/jcm10204740

**Published:** 2021-10-15

**Authors:** Tzyy-Ling Chuang, Mei-Hua Chuang, Yuh-Feng Wang, Malcolm Koo

**Affiliations:** 1Department of Nuclear Medicine, Dalin Tzu Chi Hospital, Buddhist Tzu Chi Medical Foundation, Dalin, Chiayi 622401, Taiwan; b8601139@tmu.edu.tw (T.-L.C.); yuhfeng@gmail.com (Y.-F.W.); 2School of Medicine, Tzu Chi University, Hualien 970374, Taiwan; cmh618@ms32.hinet.net; 3Faculty of Pharmacy, National Yang-Ming Chiao Tung University, Taipei City 112304, Taiwan; 4MacKay Junior College of Medicine, Nursing and Management, New Taipei City 112021, Taiwan; 5Center of Preventive Medicine, Dalin Tzu Chi Hospital, Buddhist Tzu Chi Medical Foundation, Dalin, Chiayi 622401, Taiwan; 6Graduate Institute of Long-Term Care, Tzu Chi University of Science and Technology, Hualien 970302, Taiwan

**Keywords:** bone density, trabecular bone score, normative values, dual-energy X-ray absorptiometry

## Abstract

Trabecular bone score (TBS) is a novel method for assessing trabecular microarchitecture. Normative values of TBS are available for various populations of the world but are not yet available for Taiwanese adults. Therefore, the purpose of this study was to estimate age-specific, normative TBS curves for Taiwanese men and women. Medical records of general health examinations from a regional hospital in Southern Taiwan were reviewed. Individuals aged 30–90 years with data on lumbar spine bone mineral density (BMD) were included. TBS was retrospectively calculated from dual-energy X-ray absorptiometry scans using TBS iNsight software. Of the 12,028 patients included, 4533 (37.7%) were male and the mean age was 55.8 years. The mean TBS was 1.392 (standard deviation (SD) 0.089) for men and 1.344 (SD 0.107) for women. In women, TBS declined at a rate of 0.0004/year among those aged 30.0–45.9 years, 0.0106/year among those 46.0–60.7 years, and 0.0028/year among those 60.8–90.0 years. In men, TBS declined at a constant rate of 0.0023/year over the entire age range. In conclusion, age-adjusted, normative curves of TBS for Taiwanese men and women are presented, which could be used to facilitate the use of TBS in assessing bone status in clinical practice.

## 1. Introduction

Osteoporosis is a multifactorial disorder characterized by low bone mass and deterioration of bone tissue, which can lead to increased risk of fracture. As the disease predominately occurs in postmenopausal women and older men, the social and economic burden of osteoporosis have increased steadily because of the aging of the world population [1]. It has been estimated that the prevalence of osteoporosis in industrialized countries ranged from 9 to 38% for women and 1 to 8% for men, affecting up to 49 million individuals [2].

Osteoporosis is conventionally diagnosed by areal bone mineral density (BMD) measurement using dual-energy X-ray absorptiometry (DXA). However, BMD does not always fully reflect fracture risk because the properties of bone tissue material also play a role in the ability for a whole bone to resist mechanical failure [3,4]. For instance, trabecular bone microarchitecture, a key determinant of bone strength, cannot be measured using DXA. Historically, invasive bone biopsy is the gold standard for assessing bone microarchitectural properties. A novel development in indirect measurement of the trabecular microarchitecture is the trabecular bone score (TBS), introduced in 2008 by Pothuaud et al. [5]. TBS is strongly correlated with bone microarchitecture, regardless of BMD [5,6]. It is a gray-level index of texture inhomogeneity based on the use of experimental variograms of two-dimensional projection images acquired during a standard DXA lumbar spine scan. Therefore, TBS measurement can be integrated with bone density evaluations without changing the existing workflow.

The basic principle of TBS is derived from the use of variograms in geostatistical analysis methods. The two-dimensional projection of a dense trabecular microstructure will generate an image with a large number of pixel-to-pixel gray-level variations of small amplitude. Conversely, the two-dimensional projection of a porous trabecular microstructure will produce an image with a low number of pixel-to-pixel gray-level variations of large amplitude. A variogram of the projected images, for which the sum of the squared grey-level differences between pixels at a specific distance, is calculated. TBS is computed as the slope of the log-log transform of the variogram, where the slope represents the rate of grey-level amplitude variations. A steeper variogram slope indicates better bone structure [5].

A bibliometric review of TBS indexed in the Science Citation Index Expanded and the Social Sciences Citation Index from 2008 to 2019 showed that 430 original and review articles on TBS were published during the period [7]. Several large-scale studies suggested the usefulness of TBS in addition to BMD to predict osteoporotic fractures in different population groups [7,8,9,10]. Clinically, TBS can be used in conjunction with BMD or *T*-score to improve the prediction of fracture risk, particularly in conditions wherein BMD readings lie close to the intervention threshold [11]. A meta-analysis of individual-level data from 17,809 men and women in 14 prospective population-based cohorts supported that TBS could significantly contribute to fracture prediction independent of FRAX [12]. A recent study also showed that TBS was more sensitive than BMD in detecting vertebral fracture and fragility fracture in patients with chronic inflammatory rheumatic diseases on long-term and low-dose glucocorticoid treatment [13].

For clinical use, reference standards for TBS values for different populations worldwide are required. At present, TBS normative data are available for several populations, including Australians [14], Chinese women [15], French women [16], Iranians [17], Italians [18], Japanese women [19], Mexicans [20], non-Hispanic American white women [21], Sri Lankan women [22], and Thais [23]. Nevertheless, to the best of our knowledge, TBS normative data are not available for Taiwanese adults. Therefore, the aim of this study was to develop age-adjusted normative reference values for TBS in Taiwanese men and women.

## 2. Materials and Methods

### 2.1. Study Design and Study Population

This retrospective study with data based on medical records was conducted in a regional teaching hospital in southern Taiwan. All patients who had undergone a general health examination from 1 June 2014 to 30 July 2020 were identified and reviewed. The study protocol was approved by the institutional review board of the study hospital (IRB No. B11001010). The requirement for informed consent from the patients was waived due to the use of deidentified medical records.

The exclusion criteria were patients with an age of <30 years or >90 years, lack of information on TBS or BMD, and previous history of fracture. In addition, according to the TBS software manufacturer, TBS values calculated for patients with a body mass index (BMI) < 15 kg/m^2^ or >37 kg/m^2^ may be biased. Therefore, patients with a BMI in these ranges were also excluded.

### 2.2. Demographic Data, Areal Bone Mineral Density Assessment and Trabecular Bone Score

Body weights and heights were measured by trained staff with patients in light clothing and without shoes. The data were recorded to the nearest 0.5 kg and 0.5 cm, respectively. BMI was calculated as weight in kilograms divided by the height in meters squared. It is further categorized into normal (18.5–24.0 kg/m^2^), underweight (<18.5 kg/m^2^), overweight (24.0–26.9 kg/m^2^), or obese (≥27.0 kg/m^2^) according to the criteria set by the Ministry of Health and Welfare in Taiwan.

Lumbar spine areal BMD was measured from L_1_–L_4_ using a Discovery Wi DXA system (Hologic Inc., Marlborough, MA, USA) and TBS was retrospectively calculated from L_1_–L_4_ from the same DXA scans using TBS iNsight^®^ software (version 3.0.2.0; Medimaps Group, Geneva, Switzerland). For the BMD *T*-score calculation, the manufacturer’s age-specific reference curve for Japanese women was used.

Lumbar spine areal BMD was categorized into normal, osteopenia, and osteoporosis using the World Health Organization classification [2]. A *T*-score greater than −1.0 was classified as normal, between −1.0 and −2.5 was classified as osteopenia, and less than −2.5 was classified as osteoporosis.

### 2.3. Statistical Analysis

All statistical analyses were conducted using R (version 3.6.3) [24] under the RStudio environment (version 1.4.1106) [25]. *p*-value lower than 0.05 was considered statistically significant. Results are reported as mean with standard deviation (SD) and median with upper and lower quartile (Q_1_ and Q_3_) or frequency and percentage (%), as appropriate. Pearson’s correlation coefficients were used to assess the associations between areal BMD and TBS with age.

Separate piecewise linear regression models were developed to represent age-related changes in TBS for men and women. The cut-off values of the segments were determined using the Segmented package in R [26]. Two SDs below the mean of TBS in both sexes were considered as the cutoff values of TBS for low-quality bone (degraded microarchitecture). The values between one SD and two SDs below mean TBS were considered as intermediate bone quality (partially degraded microarchitecture). The values higher than one SD below the mean of TBS were defined as normal bone quality.

## 3. Results

A total of 21,814 records of patients who underwent a general health examination from 1 June 2014 to 30 July 2020 were identified. Of patients who had undergone more than one general health examination, we randomly selected one visit from each patient, and therefore 5335 records were removed. The remaining 16,479 patients were screened for exclusion criteria, and 4451 were excluded for the following reasons: age < 30 or >90 years (*n* = 716), no TBS data (*n* = 2738), no BMD data (*n* = 98), a BMI < 15 kg/m^2^ or > 37 kg/m^2^ (*n* = 49), and previous history of fracture (*n* = 850). Therefore, 12,028 patients were included in the analysis.

The characteristics of men and women by age groups are shown in Table 1. Due to the small number of individuals aged 80 and over, the 80–84 and 85–90 years age groups were combined. Among the 12,028 individuals enrolled in the study, 4533 (37.7%) were male and the overall mean age was 55.8 years (SD = 10.7). The mean body mass index was 24.9 (SD = 3.3) and 23.5 (SD = 3.3) for men and women, respectively. According to the Taiwan BMI categories, the prevalence of underweight, normal, overweight, and obese was found to be 1.8%, 39.6%, 34.8% and 23.9% in men, and 3.8%, 57.9%, 24.1%, 14.2% in women, respectively.

The mean lumbar spine areal BMDs for men and women were 0.989 g/cm^2^ (SD = 0.150) and 0.889 g/cm^2^ (SD = 0.153), respectively. Based on the *T*-score classification of BMD for lumbar spine, 3681 (30.6%) of the patients had a normal bone density, 6357 (52.9%) had osteopenia, and 1990 (16.5%) had osteoporosis. Table 2 shows the age-specific lumbar spine areal BMD and *T*-score for men and women. The highest mean spine areal BMD was observed in men aged 35–39 years and in women aged 40–44 years. A declining trend was found only in women but not in men. In addition, areal BMD showed a significant inverse correlation with age in women (r = −0.49, 95% confidence interval (CI) −0.50, −0.47, *p* < 0.001), but not in men (r = −0.03, 95% CI −0.05, −0.01, *p* = 0.098).

Age-specific TBS for men and women are shown in Table 3. The highest TBS was observed in men aged 35–39 years and in women aged 30–34 years. A declining trend was found in both men and women. The mean TBS over the full 30–90 years age range was 1.392 (SD = 0.089) for men and 1.344 (SD = 0.107) for women. In addition, TBS showed a significant inverse correlation with age in both women (*r* = −0.62, 95% CI −0.63, −0.61, *p* < 0.001) and men (*r* = −0.30, 95% CI −0.32, −0.27, *p* < 0.001).

With TBS thresholds defined in this study with cutoff values at −1 and −2 SDs, 3829 (84.5%) of men and 6256 women (83.5%) had normal microarchitecture; 578 (12.8%) of men and 1090 women (14.5%) had partially degraded microarchitecture; and 126 (2.8%) of men and 149 women (2.0%) had degraded microarchitecture. In men, the cutoff values of −1 and −2 SDs corresponded to TBS values of 1.303 and 1.214, respectively. In women, the cutoff values of −1 and −2 SDs corresponded to TBS values of 1.237 and 1.131, respectively.

The results of piecewise linear regression analyses of TBS and age are shown in Figure 1 and Figure 2. Intermediate and low bone quality could be defined as those values below one SD (the lower dotted line) and two SDs (the lower dashed line) below the mean TBS, respectively. Moreover, in women, two breakpoints at 45.9 and 60.7 years were identified. In women, the linear regression equations were equal to; TBS = 1.463 − 0.0004 × age (years) among those 30.0–45.9 years (root-mean-square error (RMSE) = 0.075); TBS = 1.929 − 0.0106 × age (years) for those 46.0–60.7 years (RMSE = 0.083); and TBS = 1.456 − 0.0028 × age (years) for those 60.8–90.0 years (RMSE = 0.082). Conversely, no breakpoints were identified in men. The linear regression equation was equal to TBS = 1.522 − 0.0023 × age (years) for men 30.0–90.0 years (RMSE = 0.085).

## 4. Discussion

Several studies have reported normative values of TBS in women of different countries, including China [15], France [16], Japan [27], Sri Lanka [22] and the United States [21], and in both men and women of Australia [14], Iran [17], Italy [18], Mexico [20] and Thailand [23]. There is a need for establishing reference values for different populations because ethnicity is reported to be an important factor in bone quality [28] and fracture risk [29]. In this study, we reported age-specific spinal TBS for 12,028 Taiwanese men and women between the ages of 30 and 90 years. To the best of our knowledge, this is the first study to generate reference values for this population, which can be used to help clinicians when interpreting TBS results for individual patients and to serve as a basis for international comparisons.

Consistent with several previously published studies, areal BMD and TBS declined with age in women [15,27] but not in men [20]. The mean TBS over the full 30–90 year age range was 1.392 for men and 1.344 for women. Our finding in women was in-between the 1.32 reported in a study of 537 healthy Chinese women with ages from 20–89 years [15] and the 1.362 reported in population-based study of 3985 Japanese women aged 15–79 years [27]. For men, our value was higher than the 1.326 reported in a study of 381 Iranian men [17] and the 1.226 found in a study of 894 Australian men aged 24 to 98 years [14]. To put the value of TBS into perspective, based on a meta-analysis of individual-level data of TBS for fracture risk prediction on 17,809 individuals from 14 prospective cohorts in different countries or regions, a TBS > 1.31 could be considered as normal, 1.23 ≤ TBS ≤ 1.31 as partially degraded microarchitecture, and TBS < 1.23 as degraded microarchitecture in both men and women [12].

The decline in BMD and TBS with increasing age could be explained by a combination of intrinsic, such as genetics and hormonal factors, as well as extrinsic factors, such as nutrition, physical activity, and the use of certain medications. While bone resorption increases over bone formation with aging in both sexes, women experience accelerated bone loss due to low levels of estrogens in the perimenopausal period [30]. As for TBS, a study on lumbar vertebral bodies from cadavers using microcomputed tomography and scanning electron microscopy showed that trabecular number decreased whereas trabecular separation increased in older cadavers [31].

Based on the results from the piecewise linear regression model in women, two breakpoints were identified at 45.9 years and 60.7 years. Decreases of 0.0004 per year in TBS among those aged 30.0–45.9 years, 0.0106 per year among those 46.0–60.7 years, and 0.0028 per year among those 60.8–90.0 years were observed. As the mean age of onset of natural menopause was reported to be 50.2 years (SD = 4.0 years) in Taiwanese women [32], the three segments broadly corresponded to the premenopausal, perimenopausal, and postmenopausal periods. The pattern of decline in TBS with increasing age suggested that the bone microarchitecture loss occurred most rapidly among women in their mid-40s to 60s. This pattern is similar to that observed in the Japanese Population-based Osteoporosis (JPOS) study [27]. Conversely, no breakpoints were found in men based on the piecewise linear regression analysis. TBS steadily decreased at a rate of 0.0023 per year among men over the entire age span of 30–90 years. Cross-sectional data collected from a cohort of men and women assessed as part of the Geelong Osteoporosis Study in Australia also showed a constant decline of TBS with age in men. On the other hand, a best-fit model revealed an inverse cubic association between TBS and age in women [14].

Several limitations should be taken into account when interpreting the results of this study. First, our results are based on data from a cross-sectional study design. Future prospective studies are needed to confirm our cut-off points in predicting fracture risk. Second, the study population was not a nationally-representative sample, similar to a few other studies that were based on convenient sampling [18]. Despite the aforementioned limitations, this study has a few strengths. First, our sample covers both men and women over a broad adult age range. This is important as men are often understudied in osteoporosis research. Second, all our TBS measurements were conducted with the same densitometer and TBS algorithm, which eliminated potential differences in the results introduced by densitometers [33]. Third, our sample size is relatively large compared with previous studies with similar objectives.

## 5. Conclusions

In conclusion, this study provides normative TBS values for Taiwanese middle-aged and older adults. Our findings might be used to serve as a proposed reference for cross-country comparisons and to assist clinicians in assessing bone quality of their patients. Further studies are needed to confirm the validity of the cutoff points to identify the risk of future fracture.

## Figures and Tables

**Figure 1 jcm-10-04740-f001:**
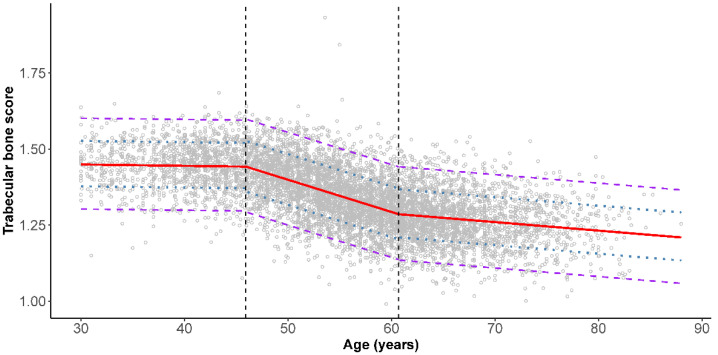
Piecewise linear regression analysis of trabecular bone score and age in female Taiwanese. Solid line (red) represents the three segments piecewise linear regression line. Two breakpoints were identified. The dotted (blue) and dashed lines (purple) represent the one and two standard deviation lines, respectively.

**Figure 2 jcm-10-04740-f002:**
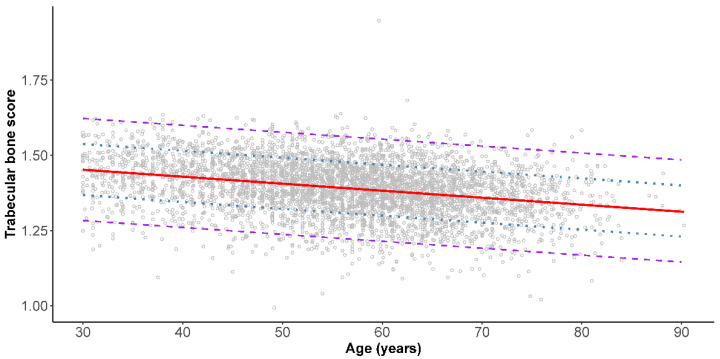
Piecewise linear regression analysis of trabecular bone score and age in male Taiwanese. Solid line (red) represents the piecewise linear regression line. No breakpoints were identified. The dotted (blue) and dashed lines (purple) represent the one and two standard deviation lines, respectively.

**Table 1 jcm-10-04740-t001:** Age-specific body mass index for Taiwanese men and women.

Age Group (Years)	Body Mass Index							
	Total		Male			Female		
	*n*	(%)	*n*	(%)	Mean (SD)	*n*	(%)	Mean (SD)
30–34	395	(3.3)	185	(4.1)	24.7 (4.4)	210	(2.8)	22.2 (3.9)
35–39	575	(4.8)	259	(5.7)	25.9 (3.9)	316	(4.2)	22.6 (3.5)
40–44	955	(7.9)	335	(7.4)	25.5 (3.6)	620	(8.3)	23.3 (3.8)
45–49	1455	(12.1)	502	(11.1)	25.3 (3.5)	953	(12.7)	23.2 (3.2)
50–54	2016	(16.8)	705	(15.6)	25.1 (3.0)	1311	(17.5)	23.3 (3.2)
55–59	2654	(22.1)	959	(21.2)	24.8 (2.9)	1695	(22.6)	23.6 (3.2)
60–64	1535	(12.8)	572	(12.6)	24.7 (2.9)	963	(12.8)	23.8 (3.3)
65–69	1336	(11.1)	521	(11.5)	24.4 (3.3)	815	(10.9)	24.0 (3.5)
70–74	705	(5.9)	303	(6.7)	24.2 (2.9)	402	(5.4)	23.8 (3.0)
75–79	316	(2.6)	146	(3.2)	23.9 (2.8)	170	(2.3)	24.2 (3.2)
80–90	86	(0.7)	46	(1.0)	23.8 (3.3)	40	(0.5)	23.9 (3.7)
Total	12,028	(100)	4533	(37.7)	24.9 (3.3)	7495	(62.3)	23.5 (3.3)

SD: standard deviation.

**Table 2 jcm-10-04740-t002:** Age-specific lumbar spine areal bone mineral density for Taiwanese men and women.

Age Group (Years)	Lumbar Spine Areal BMD				Lumbar Spine T-Score				
	Mean (SD)	Q1	Median	Q3		Mean (SD)	Q1	Median	Q3
Male									
30–34	1.000 (0.129)	0.923	1.000	1.080		−0.232 (1.093)	−0.90	−0.20	0.40
35–39	1.022 (0.133)	0.927	1.009	1.111		−0.051 (1.132)	−0.90	−0.10	0.70
40–44	1.001 (0.131)	0.917	0.991	1.086		−0.229 (1.074)	−1.00	−0.30	0.50
45–49	0.986 (0.135)	0.892	0.975	1.069		−0.376 (1.154)	−1.20	−0.40	0.40
50–54	0.985 (0.136)	0.883	0.976	1.075		−0.403 (1.170)	−1.20	−0.50	0.40
55–59	0.976 (0.150)	0.875	0.965	1.064		−0.458 (1.254)	−1.30	−0.60	0.30
60–64	0.989 (0.162)	0.883	0.975	1.088		−0.357 (1.383)	−1.30	−0.40	0.50
65–69	0.988 (0.163)	0.868	0.977	1.085		−0.353 (1.374)	−1.30	−0.40	0.50
70–74	0.983 (0.175)	0.854	0.977	1.088		−0.413 (1.497)	−1.50	−0.40	0.50
75–79	0.993 (0.172)	0.885	0.990	1.087		−0.323 (1.462)	−1.20	−0.30	0.50
80–90	1.059 (0.224)	0.926	1.011	1.230		0.220 (1.891)	−0.85	−0.15	1.48
Female									
30–34	0.995 (0.111)	0.920	0.988	1.065		−0.132 (0.978)	−0.80	−0.20	0.50
35–39	1.003 (0.117)	0.924	1.003	1.086		−0.056 (1.034)	−0.70	−0.10	0.68
40–44	1.008 (0.116)	0.933	1.003	1.083		−0.005 (1.018)	−0.70	0.00	0.60
45–49	0.992 (0.133)	0.904	0.988	1.080		−0.158 (1.170)	−0.90	−0.20	0.60
50–54	0.920 (0.139)	0.819	0.916	1.013		−0.789 (1.219)	−1.70	−0.80	0.00
55–59	0.847 (0.128)	0.754	0.837	0.925		−1.420 (1.129)	−2.20	−1.50	−0.70
60–64	0.816 (0.135)	0.718	0.804	0.902		−1.689 (1.187)	−2.60	−1.80	−1.00
65–69	0.809 (0.138)	0.717	0.799	0.892		−1.737 (1.200)	−2.50	−1.80	−1.00
70–74	0.790 (0.132)	0.694	0.776	0.882		−1.924 (1.164)	−2.80	−2.00	−1.20
75–79	0.792 (0.131)	0.697	0.785	0.868		−1.899 (1.145)	−2.70	−2.00	−1.20
80–90	0.758 (0.165)	0.656	0.713	0.825		−2.220 (1.449)	−3.08	−2.60	−1.63

BMD: body mineral density; Q_1_: first quartile; Q_3_: third quartile; SD: standard deviation.

**Table 3 jcm-10-04740-t003:** Age-specific trabecular bone score for Taiwanese men and women.

Age Group (Years)	Trabecular Bone Score							
	Male				Female			
	Mean (SD)	Q1	Median	Q3	Mean (SD)	Q1	Median	Q3
30–34	1.438 (0.086)	1.392	1.439	1.498	1.448 (0.072)	1.401	1.451	1.495
35–39	1.440 (0.087)	1.386	1.447	1.500	1.444 (0.075)	1.398	1.449	1.497
40–44	1.427 (0.081)	1.372	1.435	1.482	1.446 (0.073)	1.399	1.449	1.496
45–49	1.412 (0.084)	1.362	1.418	1.469	1.423 (0.082)	1.374	1.429	1.478
50–54	1.400 (0.083)	1.348	1.408	1.456	1.375 (0.087)	1.317	1.375	1.436
55–59	1.388 (0.086)	1.333	1.390	1.446	1.314 (0.084)	1.260	1.313	1.368
60–64	1.375 (0.086)	1.324	1.380	1.427	1.282 (0.081)	1.232	1.282	1.331
65–69	1.365 (0.082)	1.315	1.369	1.421	1.265 (0.082)	1.211	1.264	1.314
70–74	1.347 (0.086)	1.298	1.347	1.411	1.252 (0.083)	1.197	1.248	1.307
75–79	1.349 (0.087)	1.297	1.348	1.403	1.246 (0.082)	1.189	1.248	1.305
80–90	1.344 (0.095)	1.282	1.356	1.413	1.225 (0.084)	1.154	1.237	1.287
Total	1.392 (0.089)	1.335	1.396	1.453	1.344 (0.107)	1.267	1.342	1.424

Q_1_: first quartile; Q_3_: third quartile; SD: standard deviation.

## Data Availability

The data used to support the findings of this study are available from the corresponding author upon request.

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
