# Peer review of "Age-Specific Normative Values of Lumbar Spine Trabecular Bone Score (TBS) in Taiwanese Men and Women"

_jcm, 2021, doi:10.3390/jcm10204740_

Round 1

Reviewer 1 Report

Cross-sectional data base study on TBS in a Taiwanese population. It remains unclear, why DXA-measurement had been performed in this population. To receive data to create normal values healthy persons have to be identified and tested. There is no data concerning the medical history relevance for inclusion / exclusion.

Have  there been exclusions at all, and if so, what were the reasons and the numbers?

Author Response

Reviewer 1, Comment 1:

Cross-sectional data base study on TBS in a Taiwanese population. It remains unclear, why DXA-measurement had been performed in this population. To receive data to create normal values healthy persons have to be identified and tested. There is no data concerning the medical history relevance for inclusion / exclusion.

Response to Reviewer 1, Comment 1:

We thank the reviewer for taking the time to review our manuscript and provided us with positive and valuable comments. Our study sample was assembled using the database of individuals who had undergone a general health examination in our study hospital. DXA is routinely performed procedure in our general health examination package. The exclusion criteria were patients with an age of < 30 years or > 90 years, lack of information on TBS or BMD, and previous history of fracture. Patients with a BMI < 15 kg/m2 or > 37 kg/m2 were also excluded based on the recommendation from the TBS software manufacturer (Line 91-95).

Reviewer 1, Comment 2:

Have there been exclusions at all, and if so, what were the reasons and the numbers?

Response to Reviewer 1, Comment 2:

We thank the reviewer for the comment. A total of 21,814 records of patients who underwent a general health examination from June 1, 2014 to July 30, 2020 were identified. Because some patients had undergone more than one general health examination during the observational period, we randomly selected one visit from each patient, and therefore 5,335 records were removed.

The remaining 16,479 patients were screened for exclusion criteria, and 4,451 were excluded for the following reasons: age < 30 or > 90 years (n = 716), no TBS data (n = 2,738), no BMD data (n = 98), a BMI < 15 kg/m2 or > 37 kg/m2 (n = 49), and previous history of fracture (n = 850). Therefore, 12,028 patients were included in the analysis (Line 127-134).

Reviewer 2 Report

Dear Authors, your work is very interesting and different sections are well balanced. Your conclusions are supported by the results

Author Response

We greatly appreciate the reviewer for taking the time to review our manuscript.

Reviewer 3 Report

Thank you for allowing me to review this review article entitled: "age-specific normative values of lumbar spine trabecular bone score (TBS) in Taiwanese men and women."

This article is well written, and this topic may be valuable to inform a reference standard for TBS values in Taiwanese people.

However, I have some points before considering publication.

1. What is the novelty or significant difference of your manuscript compared to previously published studies?

2. There is a lack of an extensive comparison between TBS and BMD (bone mineral density). It might be helpful to discuss the theoretical or clinical advantages of the application of TBS compared to BMD.

Author Response

Reviewer 3, Comment 1:

This article is well written, and this topic may be valuable to inform a reference standard for TBS values in Taiwanese people.

However, I have some points before considering publication.

What is the novelty or significant difference of your manuscript compared to previously published studies?

Response to Reviewer 3, Comment 1:

We appreciate the comment from the reviewer. First, while TBS normative data are available for several populations, many included only women. Our study included both sexes and covered a broad age range. Second, our sample size is relatively large compared with previous studies with similar study objectives. Third, we fitted our data using piecewise linear regression models that incorporate breakpoints to allow different relationships over the broad age range. Fourth, this is the first study that generated reference values for the Taiwanese population, which can be used to help clinicians when interpreting TBS results for local patients and to serve as a basis for international comparisons.

-----------------------------------------------------------------

Reviewer 3, Comment 2:

There is a lack of an extensive comparison between TBS and BMD (bone mineral density). It might be helpful to discuss the theoretical or clinical advantages of the application of TBS compared to BMD.

Response to Reviewer 3, Comment 2:

We appreciate the comment from the reviewer. We have added a new paragraph in the Introduction to describe the principle of TBS measurement (Line 50-61). We have also added the reference of a meta-analysis supporting that TBS can predict fracture independent of BMD (Line 68-70) and a new study showing that TBS was more sensitive than BMD in detecting vertebral fracture and fragility fracture in patients with chronic inflammatory rheumatic diseases on long-term and low-dose glucocorticoid treatment (Line 71-73).